# Comparative Genome Analysis of Japanese Field-Isolated *Aspergillus* for Aflatoxin Productivity and Non-Productivity

**DOI:** 10.3390/jof10070459

**Published:** 2024-06-28

**Authors:** Tomohiro Furukawa, Kanae Sakai, Tadahiro Suzuki, Takumi Tanaka, Masayo Kushiro, Ken-Ichi Kusumoto

**Affiliations:** 1Institute of Food Research, National Agriculture and Food Research Organization (NARO), 2-1-12 Kannondai, Tsukuba 305-8642, Japan; 2Department of Biotechnology, Graduate School of Engineering, Osaka University, 2-1 Yamadaoka, Suita 565-0871, Japan

**Keywords:** aflatoxin, *Aspergillus flavus*, *Aspergillus oryzae*, biosynthetic gene cluster, genetic variants, gene ontology enrichment analysis

## Abstract

*Aspergillus flavus* produces aflatoxin, a carcinogenic fungal toxin that poses a threat to the agricultural and food industries. There is a concern that the distribution of aflatoxin-producing *A. flavus* is expanding in Japan due to climate change, and it is necessary to understand what types of strains inhabit. In this study, we sequenced the genomes of four *Aspergillus* strains isolated from agricultural fields in the Ibaraki prefecture of Japan and identified their genetic variants. Phylogenetic analysis based on single-nucleotide variants revealed that the two aflatoxin-producing strains were closely related to *A. flavus* NRRL3357, whereas the two non-producing strains were closely related to the RIB40 strain of *Aspergillus oryzae*, a fungus widely used in the Japanese fermentation industry. A detailed analysis of the variants in the aflatoxin biosynthetic gene cluster showed that the two aflatoxin-producing strains belonged to different morphotype lineages. RT-qPCR results indicated that the expression of aflatoxin biosynthetic genes was consistent with aflatoxin production in the two aflatoxin-producing strains, whereas the two non-producing strains expressed most of the aflatoxin biosynthetic genes, unlike common knowledge in *A. oryzae*, suggesting that the lack of aflatoxin production was attributed to genes outside of the aflatoxin biosynthetic gene cluster in these strains.

## 1. Introduction

Aflatoxins (AFs) are potent toxins produced by some *Aspergillus* fungi as secondary metabolites [1]. Among AFs, aflatoxin B_1_ (AFB_1_) is the strongest known carcinogen. AF-producing fungi include *Aspergillus flavus*, *A. parasiticus*, the less common *A. nomius*, *A. pseudotamarii*, *A. bombycis*, and *A. parvisclerotigenus*. Among these species, *A. flavus* is considered the most problematic because it infects major crops such as corn, cotton, peanuts, and tree nuts worldwide and contaminates them with AFB_1_ [2]. AF contamination not only harms humans and livestock, but also causes significant economic losses due to the disposal of contaminated agricultural products [3]. *A. flavus* resides in agricultural soil in its habitat, is carried by wind or insects to attach to crops, proliferates during crop storage, and accumulates AFB_1_ [4]. AF is biosynthesized in a multi-enzymatic process consisting of at least 18 steps from acetyl-CoA as a starter unit and Sterigmatocystin (ST) as a penultimate precursor [5] (Figure 1). Genes encoding AF biosynthetic enzymes and transcription factors that regulate the expression of biosynthetic enzyme genes are clustered in approximately 70 kb of the chromosome, named the AF biosynthetic gene cluster (BGC) [6]. AF-producing and non-producing strains of *A. flavus* exist, and it is difficult to distinguish between them based on the appearance of the fungus on agar plates without specific additives to detect AF production [7]. DNA barcode regions, such as the nuclear ribosomal internal transcribed spacer (ITS) region, are also identical; therefore, PCR analysis using universal primers does not work [8]. Furthermore, *Aspergillus oryzae*, which forms colonies like *A. flavus* but does not produce AFs, is widely used in the brewing industry to produce traditional Japanese foods such as sake, soy sauce, and miso. *A. flavus* and *A. oryzae* have 99.5% nucleotide similarity genomes (genome sizes of 36.8 and 36.7 Mb, respectively, and 12,197 and 12,079 genes, respectively) and cannot be distinguished in the DNA barcode region [9]. *A. flavus* and *A. oryzae* are included in *Aspergillus* section *Flavi*, which share the common characteristic of yellowish-green to brownish conidial head color. *A. oryzae* is recognized as a domesticated fungus that originated from the section *Flavi*, chosen for its saccharification ability suitable for brewing and the lack of AF production [10,11].

To estimate the potential risk of AF contamination in local agricultural products, it is important to investigate the geological distribution of AF-producing strains in field soils. In Japan, an extensive survey in the 1970s concluded that AF-producing *A. flavus* inhabits only areas with an average annual temperature of 16 °C or higher and does not grow except in the Kyusyu region [12]. However, subsequent studies reported the isolation of AF-producing *A. flavus* from Kanagawa and Ibaraki prefectures, located in the Kanto region, suggesting that AF-producing strains are widely distributed in Japan [13]. In fact, according to the Japan Meteorological Agency, the long-term trend of annual average temperatures in Japan and Ibaraki have been increasing at a rate of 1.35 °C and 2.3 °C per 100 years, respectively; in 2023, the annual average temperature reached 16.1 °C in Tsukuba, Ibaraki. In 2020, we isolated four strains that were identified as *A. flavus* based on colony morphology and DNA barcode regions from the field soil in Ibaraki [14]. Two strains were AF-producing, whereas the other two strains did not produce AF. To date, no study has analyzed the genomes of *A. flavus* isolated from Japanese soils. A genomic analysis of field isolates is expected to reveal the phylogenic characteristics of the strains in the area and provide clues to the novel factors that discriminate between AF-producing and non-producing strains. In this study, we analyzed the genomes of these four strains and detected genetic variants to identify the factors responsible for the differences in AF production and quantified the gene expression of AF BGC.

## 2. Materials and Methods

### 2.1. Fungal Strains and Culture Conditions

The four putative *Aspergillus* strains used in this study were isolated from the soil of a sorghum field in Tsukuba, Ibaraki, Japan, in July 2020 and September 2020; they were designated JUL1 [14], JUL10 [14], SEP1, and SEP5. The numbers indicate the number of isolated fungal colonies. In detail, 0.15 g of soil 5 cm below the surface was collected and diluted in 500 μL of 0.05% tween-20, then spread on YES agar plates (20 g yeast extract (Difco, Sparks, MD, USA), 20 g agar, 100 g sucrose, 1 g sodium deoxycholate, and 0.1 g chloramphenicol per liter) and incubated at 25 °C in the dark for 7 days. The colonies that were visually similar to *A. flavus* were isolated. From these isolates, the sequencing of the ITS regions and partial calmodulin gene allowed the identification of four strains as *A. flavus*.

JUL1, JUL10, SEP1, and SEP5 were deposited in the GenBank project of the National Agriculture and Food Research Organization, and their registration numbers are MAFF 111211, MAFF 111212, MAFF 111223, and MAFF 111224, respectively (https://www.gene.affrc.go.jp/index_ja.php; last accessed on 12 April 2024). *A. flavus* NRRL3357 strain was obtained from the Medical Mycology Research Center, Chiba University, Chiba, Japan, through the National BioResource Project (http://www.pf.chiba-u.jp/bioresoures/index.html; last accessed on 12 April 2024). The conidia of these strains were collected from a 1-week-old culture on 2% potato dextrose agar (PDA; Difco), suspended in 30% glycerol solution, and stored at −80 °C. For the observation of colony morphology, 1 μL of 5 × 10^4^ conidia/μL suspension was inoculated on DG18 (31.5 g Dichliran-Glycerol (DG18) Agar Base (Oxoid Limited, Waltham, MA, USA), 220 g glycerol, 0.01 g ZnSO_4_·7H_2_O, 0.005 g CuSO_4_·7H_2_O, 0.05 g chloramphenicol, 0.05 g chlortetracycline per liter, pH 5.6), and MEA (30 g malt extract agar (Difco), 10 g agar, 20 g glucose, 1 g peptone, 0.01 g ZnSO_4_·7H_2_O, 0.005 g CuSO_4_·7H_2_O per liter) agar plates, and incubated at 28 °C in the dark for 7 days. For AFB_1_ analysis, potato dextrose broth (PDB; Difco) was inoculated with the conidia at 5 × 10^4^ conidia/mL in a 12-well microplate (2 mL/well), and the microplate was placed at 28 °C in the dark for 24, 36, or 48 h. The culture broth was transferred to microtubes and centrifuged to separate the mycelia from the supernatant. The mycelia were washed with distilled water, lyophilized, and weighed to determine the dry weight. The supernatants were then subjected to an AFB_1_ analysis.

### 2.2. Quantification of AFB_1_ Production

To extract AFB_1_, the culture supernatant (0.5 mL) was mixed with an equal amount of chloroform, and the chloroform layer was collected and evaporated. The remaining residue was dissolved in 90% aqueous acetonitrile (0.5 mL) and subjected to liquid chromatography/mass spectrometry (LC-MS) as previously described [15]. The LC-MS conditions were as follows: LC: Accela pump (Thermo Fisher Scientific, Waltham, MA, USA); column: CAPCELL PAK C_18_ MGIII (150 × 4.6 mm, 3 μm; Osaka Soda, Osaka, Japan); solvent: 0.1% formic acid (A) and acetonitrile (B); elution: a linear gradient of 5–95% B to 22 min; flow rate: 0.45 mL/min; the retention time of AFB_1_: 15.9 min; MS: Orbitrap Exactive (Thermo Fisher Scientific); ionization: electrospray ionization in positive mode; spray parameters: sheath gas/aux gas/sweep gas, 30/5/0 arbitrary units; capillary temperature/heater temperature, 250 °C/250 °C; spray voltage, 4.0 kV. For peak area quantification, the *m*/*z* range of 313.0691–313.0723 was extracted as the [M + H]^+^ of AFB_1_ (formula: C_17_H_12_O_6_). Calibration curves were determined using an aflatoxin mixture standard (containing 25 μg/mL B_1_, B_2_, G_1_, and G_2_; FUJIFILM Wako Pure Chemical, Osaka, Japan).

### 2.3. Sequencing and Assembly of the Genome of Four Strains

After 48 h of growth in PDB, the mycelia of JUL1, JUL10, SEP1, and SEP5 were harvested and lyophilized. The dry mycelia were ground to a powder in liquid N_2_ using a mortar and pestle, and genomic DNA was extracted using NucleoBond HMW DNA (TaKaRa, Shiga, Japan). The prepared DNA was shipped to the Bioengineering Laboratory, Sagamihara, Japan. Library preparation, quality check, the preparation of single-stranded circular DNA library, and DNA nanoball (DNB) preparation were performed using the MGIEasy FS DNA Library Prep Set (MGI Tech, Shenzhen, China), Agilent 2100 Bioanalyzer (Agilent Technologies, Santa Clara, CA, USA), MGIEasy Circularization Kit (MGI Tech), and DNBSEQ-G400RS High-throughput Sequencing Kit (MGI Tech), respectively. The sequencing was performed using a DNBSEQ-G400 (MGI Tech) generating paired-end 200 bp reads. The raw reads were assembled using MaSuRCA (v3.4.2), with an average 53X sequence coverage. The integrity of the assembled genome was verified using BUSCO (v5.4.4_c1) with Eukaryota as the model (255 single-copy orthologous genes). The raw reads are available at the Sequence Read Archive (DRA accession numbers: DRR528608, DRR528609, DRR528610, and DRR528611; https://www.ncbi.nlm.nih.gov/sra; last accessed on 12 April 2024).

### 2.4. Comparative Genome Analyses

The reference genome used in this study for *A. flavus* NRRL3357 was genome assembly JCVI-afl1-v3.0 (accession: GCA_000006275.3). The assembled JUL1, JUL10, SEP1, and SEP5 genomes were aligned to the reference genome using the Mashmap aligner (v2.0) in the D-GENIES tool [16]. The raw sequence reads were mapped to the reference sequence according to the pipeline shown in Appendix A. The pipeline contains verified tools to perform read quality assessment, alignment, variant identification, variant annotation, and visualization required for the variant analysis of NGS sequencing data [17]. In addition to these four strains, mapping was performed on the raw sequence reads of *A. oryzae* RIB40, *A. flavus* AF70, *A. parasiticus* CBS 117618, and *A. minisclerotigenes* CBS 117635, downloaded from the Sequence Read Archive (accession: SRR1835311, SRR6659155, SRR8840397, and SRR8398929. https://www.ncbi.nlm.nih.gov/sra, last accessed on 14 June 2024). First, the low-quality bases and adapter sequences were removed using Trimmomatic (v0.39) [18]. The reference genome was indexed, and the reads were mapped to the indexed reference using bwa (v0.7.17) [19]. Duplicate reads were removed using Picard (v2.27.5) [20]. Variant sites, including single-nucleotide variants (SNVs) and indels (insertions and deletions), were identified by gatk (v4.3.0.0) [21] Filtering conditions for SNVs: QD < 2.0, QUAL < 30.0, SOR > 3.0, FS > 60.0, MQ < 40.0, MQRankSum < −12.5, ReadPosRankSum < −8.0, and filtering conditions for indels: QD < 2.0, QUAL < 30.0, FS > 200.0.

A phylogenetic tree of the SNVs detected among the nine species and strains (JUL1, JUL10, SEP1, SEP5, *A. flavus* NRRL3357, *A. oryzae* RIB40, *A. flavus* AF70, *A. parasiticus* CBS 117618, and *A. minisclerotigenes* CBS 117635) was constructed using RAxML-NG (v1.2.1) with 1000 bootstrap replicates using TVM as a model [22]. The tree was visualized using FigTree (v1.4.4) [23].

The effects of variants in JUL1, JUL10, SEP1, and SEP5 were annotated and predicted using snpEff (v5.0e) [24]. The genome annotation database required to run snpEff was constructed using an annotation file (GFF format) downloaded from https://www.ncbi.nlm.nih.gov/datasets/genome/GCA_000006275.3/ (accessed on 12 April 2024). Mapping and variant detection results were visualized using the Integrative Genomics Viewer (IGV) [25].

The gene ontology (GO) enrichment analysis of the genes with impact HIGH variants (predicted by snpEff) unique to each strain was performed using the GO enrichment tool in FungiDB [26]. The summarization and visualization of biological process terms were performed using REVIGO [27]. The scatter plots generated as REVIGO outputs were visualized using Rstudio (v2023.06.0) and R (v4.3.1).

### 2.5. Gene Expression Analysis by RT-qPCR

Candidate primers for AF BGC genes were designed using the Primer Express software (v3.0; Thermo Fisher Scientific). The sequence of each gene targeted for primer design was based on the annotation of the *A. flavus* NRRL3357 genome assembly JCVI-afl1-v3.0. Primer positions were checked using IGV, and the primers recognizing 100% matching sequences among NRRL3357, JUL1, JUL10, SEP1, and SEP5 were used. The primer sequences are listed in Appendix A. To prepare cDNA, the lyophilized mycelia of each strain were ground under liquid N_2_ using a mortar and pestle. Total RNA was extracted using TRIzol reagent (Thermo Fisher Scientific) and purified using a PureLink RNA Mini Kit (Thermo Fisher Scientific). On-column DNase I treatment was performed during purification according to the manufacturer’s instructions. Complementary DNA was synthesized using ReverTra Ace qPCR RT Master Mix (TOYOBO, Osaka, Japan). Quantitative PCR was conducted using the THUNDERBIRD Next SYBR qPCR Mix (TOYOBO) in a QuantStudio 12 K Flex Real-Time PCR system (Thermo Fisher Scientific). No amplification was observed in the non-reverse transcription control. The mRNA levels for each gene calculated by the relative quantification method were normalized to those of control *β-tubulin* genes (forward primer: 5′-AGCTCTCCAACCCCTCTTACG-3′ and reverse primer: 5′-TGAGCTGACCGGGGAAACG-3′) for each sample. The mRNA levels of each gene were standardized such that the average value was 0 and the variance was one in all the samples. Using these standardized values, a principal component analysis (PCA) was performed using the ggbiplot package (v0.6.2) in R (v4.3.1).

### 2.6. Statistical Analysis and Data Visualization

The data are presented as mean ± standard deviation in bar graphs. The quantitative experiments were performed using three (sterigmatocystin (ST) feeding) or four (time-course AF and RT-qPCR) biological replicates (*n* = 3 or 4). In the ST feeding experiment, significant differences between the control and addition groups were tested using multiple unpaired *t*-tests, followed by a two-stage step-up procedure to control the false discovery rate at 0.1. For RT-qPCR, significant differences among the strains were determined using Tukey’s multiple comparison test and visualized using the compact letter display method. All the statistical tests were performed using GraphPad Prism (v10.2.1; GraphPad Software, San Diego, CA, USA). The statistical significance was set at *p* < 0.05. The Venn diagrams, bar graphs, and plots were generated using R-Studio (v2023.06.0) and R (v4.3.1).

## 3. Results

### 3.1. JUL10 Produced More AFB_1_ Than JUL1 and NRRL3357, but SEP1 and SEP5 Did Not Produce AFB_1_

In 2020, four putative *Aspergillus* strains were isolated from a field soil in Ibaraki, Japan, and named JUL1, JUL10, SEP1, and SEP5. These four strains were judged to be *A. flavus* based on colony morphology and DNA barcode region. JUL1, JUL10, SEP1, and SEP5 were grown on DG18 and MEA plates at 28 °C for 7 days (Figure 1a). The growth of these strains was similar on both media, with JUL1 and JUL10 showing a slightly darker green color. The AFB_1_ production and mycelial growth of these strains and the *A. flavus* type strain NRRL3357 were examined after 24, 36, and 48 h of incubation in PDB (Figure 1b). In NRRL3357, JUL1 and JUL10, AFB_1_ were detected after merely 24 h. Within the following 12 h, a relatively strong increase in AFB_1_ production was observed for the two isolates JUL1 and JUL10 in contrast to NRRL3357. After 48 h, AFB_1_ accumulation was 12 ng/mL for NRRL3357, 206 ng/mL for JUL1, and 792 ng/mL for JUL10. Fungal growth was similar for all five strains (Figure 1c).

### 3.2. JUL1 and JUL10 Are Phylogenetically Close to A. flavus NRRL3357 While SEP1 and SEP5 to A. oryzae RIB40

Appendix A shows an overview of the pipeline, from genomic DNA preparation to the detection of genetic variants and phylogenetic analysis. Genomic DNA was extracted from the mycelia of JUL1, JUL10, SEP1, and SEP5; whole-genome sequencing was performed; and the obtained raw reads were assembled into contigs. Assembly quality was measured using BUSCO [28], and the completeness scores were 96.9%, 95.7%, 96.5%, and 95.7% for JUL1, JUL10, SEP1, and SEP5, respectively, indicating that sufficient sequencing reads were obtained for genome assembly. The dot plots of the assembled contigs versus the reference genome of *A. flavus* NRRL3357 (genome assembly JCVI-afl1-v3.0) showed that each contig was properly aligned against the *A. flavus* genome without large gaps (Appendix A).

The sequence raw reads were mapped to the *A. flavus* NRRL3357 reference genome. The percentages of the properly mapped paired reads were 96.3%, 96.1%, 95.6%, and 95.2% for JUL1, JUL10, SEP1, and SEP5, respectively. To determine the phylogenetic position of these four strains, the sequencing reads of *A. oryzae* RIB40, *A. flavus* AF70, *A. parasiticus* CBS 117618, *A. minisclerotigenes* CBS 117635 were also mapped to the NRRL3357 reference. *A. minisclerotigenes* has recently been reported to be closer relatives of *A. oryzae* than *A. flavus* [10]. From the mapping results, variant calling was performed to detect short variants, such as SNVs and short insertions/deletions (indels). A total of 2,216,644 SNVs were detected from these nine species and strains, and a phylogenetic tree was constructed using maximum likelihood estimation (Figure 2). The tree shows that while JUL1 and JUL10 were close to the AF-producing strain *A. flavus* NRRL3357, SEP1 and SEP5 were close to the AF-non-producing strain *A. oryzae* RIB40. The evolutionary distance between SEP1 and SEP5 was minimal, indicating few differences in the genomic sequences. The JUL and SEP groups were apart, suggesting that the JULs were derived from *A. flavus* and the SEPs from the *A. oryzae* group.

### 3.3. Gene Ontology (GO) Enrichment Analysis Suggests Different Biological Processes Are Impaired among JUL1, JUL10, and SEPs

SnpEff can predict the impact of detected gene variants on gene function and classify them into several categories [22]. Each variant was classified as HIGH, MODERATE, LOW, or MODIFIER based on its putative impact on gene function. For example, the variants assumed to have function-disruptive impact, such as stop-gain (nonsense), frameshift, and intron variations, are classified as HIGH. A non-disruptive variant that might change protein effectiveness, such as missense mutations and inframe codon deletions, are classified as MODERATE. Depending on location, the MODERATE variants can be fatal to protein function, but are not judged by SnpEff. The variants unlikely to change protein function, such as synonymous variant (codon change that produces the same amino acid) are classified as LOW. The variants in the non-coding region where there is no evidence of impact, such as the variants downstream of a gene, are classified as MODIFIER.

The genetic variants detected in the JULs and SEPs against the *A. flavus* NRRL3357 reference were subjected to SnpEff, and the lists of putative variant effects were obtained (Appendix A). Because HIGH impact variants are considered to cause disruptive changes in gene function, the genes with HIGH impact variants exclusively found in each strain may reflect different characteristics among the strains. Therefore, only the genes with HIGH impact variants detected in each strain were examined. A total of 1597 and 1593 genes with HIGH impact variants were detected in SEP1 and SEP5, respectively, of which 1579 were common, consistent with their phylogenetic relatedness (Figure 3a). Hence, the lists of SEP1 and SEP5 genes with HIGH impact variants were combined into “SEPs” and it was compared with the JUL1 and JUL10 lists. A total of 241, 245, and 535 genes were exclusively found in JUL1, JUL10, and SEPs, respectively (Figure 3b).

The gene ontology (GO) enrichment analysis of these genes was performed to determine which biological processes (BP) were affected in each strain. The scatter plots of the enriched GO terms for JUL1 (Figure 3c), JUL10 (Figure 3d), and SEPs (Figure 3e) are shown. In JUL1, sulfur compound metabolic processes, such as homoserine/homocysteine conversion and *S*-adenosylmethionine metabolism, were enriched with low *p*-values. This is caused by the HIGH impact variants in the cystathionine gamma-synthase (AFLA_005895) and *S*-adenosylmethionine synthase (AFLA_004801) genes, both of which are involved in sulfur compound metabolic processes. In JUL10, the glycerophospholipid catabolic process and glycerolipid catabolic process were enriched with low *p*-values owing to the HIGH impact frameshift variants in the glycerophosphocholine phosphodiesterase gene (AFLA_008257), which belong to these two GO terms. In addition, protein phosphorylation was enriched in the JUL10 cells. In SEPs, glycerol-3-phosphate and alditol phosphate metabolic processes were enriched, as the HIGH impact intron retention variant was detected in the glycerol-3-phosphate dehydrogenase gene (AFLA_008273), which belongs to these two GO terms.

### 3.4. Disruptive Genetic Variants Were Detected for aflW (moxY), aflT, and aflO (omtB) in AF Biosynthetic Gene Cluster Only in SEP1 and SEP5

Considering that the differences in AF productivity may be due to the genetic variants within the AF biosynthetic gene cluster (BGC) (Figure 4a), the number of HIGH and MODERATE impact genetic variants in AF BGC was counted for each strain (Table 1). In JUL1 and JUL10, impact HIGH variants were detected only in *aflYa* (*nadA*) (Appendix A), *aflO* (*omtB*) (Figure 4e), *aflLa* (*hypB*) (Appendix A), *aflL* (*verB*) (Appendix A), and *aflU* (*cypA*) (Figure 4f) in JUL1. In SEP1 and SEP5, in addition to the genes detected in JULs, impact HIGH variants were detected in *aflW* (*moxY*) (Figure 4c) and *aflT* (Figure 4d). Based on the putative functions of these genes in the AF biosynthesis pathway (Figure 1), we determined the effects of these genetic variants on AF productivity. Furthermore, detailed sequence changes in *aflR*, a gene encoding a transcriptional regulator of AF biosynthetic enzyme genes, and *aflS* (*aflJ*), a gene encoding a putative regulator that interacts with AflR, were analyzed (Figure 4b) [29,30].

No HIGH variants were observed in the *aflR*-*aflS* (*aflJ*) region; however, several MODERATE variants with missense mutations were detected (Figure 4b). In the *A. oryzae* type strain RIB40, several nucleotide substitutions are known in the *aflR* and *aflS* (*aflJ*) genes and in the shared promoter region of both genes compared to *A. flavus* NRRL3357 (Figure 4b, red letters) [32]: six nucleotide substitutions in the promoter region and two and four amino acid mutations in *aflR* and *aflS* (*aflJ*), respectively. Because these nucleotide and amino acid substitutions are conserved in a group of *A. oryzae*, including the RIB40 strain, these sites are available for discrimination between *A. oryzae* and *A. flavus* [32]. SEP1 and SEP5 had conserved mutations in the RIB40 strain group (Figure 4b, red letters) and were therefore classified as *A. oryzae* rather than *A. flavus* based on this region. However, SEP1 and SEP5 also have other nucleotide substitutions in the promoter region and amino acid substitutions in *aflR* and *aflS* (*aflJ*), which have not been reported in *A. oryzae* RIB40. For JUL1 and JUL10, the shared promoter region was identical to that of *A. flavus* NRRL3357, and three missense mutations were detected in *aflS* (*aflJ*) in both JUL1 and JUL10 and one missense variant in *aflR* in JUL10 (Figure 4b).

In *aflW* (*moxY*), which encodes hydroxyversicolorone (HVN) monooxygenase and is involved in the conversion of HVN to versiconal hemiacetal acetate (VHA) [33], stop-gain variants were found in SEP1 and SEP5 (Figure 4c). A domain search indicated that the loss of residue 467 may impair the binding of NADPH, a cofactor of HVN monooxygenase (Appendix A).

In SEP1 and SEP5, a 258 bp deletion was observed in the *aflT* gene (Figure 4d), which was consistent with the 257 bp deletion conserved in the *A. oryzae* RIB40 group, supporting the classification of SEP1 and SEP5 as *A. oryzae* [34]. Frameshift mutations were detected in SEP1 and SEP5, consistent with *A. oryzae* RIB40. The domain search predicted that these mutations would result in the loss of the latter two of the 15 putative transmembrane helices of AflT protein (Appendix A). The HIGH variants in JUL1 and JUL10 had no effect on *aflT*.

*aflYa* (*nadA*) is presumed to encode the enzyme involved in the final step of AFG_1_ and AFG_2_ biosynthesis together with the enzymes encoded by *aflF* (*norB*) and *aflU* (*cypA*) [35]. In *aflYa* (*nadA*), stop-gain variants were detected in JULs and SEPs at codon 99 and 373, respectively (Appendix A). Similar to *aflF* (*norB*) and *aflU* (*cypA*) described below, *aflYa* (*nadA*) may not function properly in the four strains, even if expressed, which contributed to the non-production of G-type AF in JUL1 and JUL10.

The *aflO* (*omtB*) gene is predicted to encode *O*-methyltransferase I, the enzyme involved in the conversion of demethylsterigmatocystin (DMST) to ST and dihydrodemethylsterigmatocystin (DHDMST) to dihydrosterigmatocystin (DHST) [36,37]. In *aflO* (*omtB*), an intron variant that increased the number of second exons by 50 bases was detected in SEP1 and SEP5 (Figure 4e). The HIGH impact variants commonly detected by snpEff in JUL1, JUL10, and SEP1 were merely equivalent to 2 amino acid substitutions (Figure 4e). If these amino substitutions, including other locations, do not alter *aflO* (*omtB*) function, *aflO* (*omtB*) is considered to act normally in JUL1 and JUL10, but not in SEP1 and SEP5.

*aflLa* (*hypB*) is presumed to encode the enzyme catalyzing the oxidation of 11-hydroxy-*O*-methylstergmatocystin (HOMST) in the complex steps from *O*-methylsterigmatocystin (OMST) to AF [38]. In JUL1, JUL10, SEP1, and SEP5, a 107 bp deletion was found upstream of the gene. In SEP1 and SEP5, frameshift variants were detected near the 3′ end region of the gene (Appendix A).

*aflL* (*verB*) encodes a putative P450 monooxygenase/desaturase involved in the conversion of versicolorin B (VB) to VA [39]. Because AFB_1_ and AFG_1_ are produced from the VA, whereas AFB_2_ and AFG_2_ are produced from the VB, *aflL* (*verB*) may be involved in the divergence of 1 group AF and 2 group AF. JUL1, JUL10, SEP1, and SEP5 shared a frameshift variant at codon 19 (Proline 19 frameshift) (Appendix A), which may impair *aflL* (*verB*) function. Therefore, the presence of a desaturase that complements the AflL (VerB) function is assumed in JUL1 and JUL10.

*aflU* (*cypA*), encoding a putative cytochrome P450 monooxygenase, is involved in the divergence from the B-type AF to the G-type AF pathway, and it has been suggested that *A. flavus* produces only B-type AF owing to the loss of the AflU (CypA) function [31]. The structure of the *aflU* (*cypA*)–*aflF* (*norB*) region has been reported to correlate with AF production levels and sclerotial morphology in some strains of *A. flavus* [31,40]. *A. flavus* NRRL3357 is a large sclerotial type (L-strain) and is thought to produce less AF than small sclerotial type (S-strain) such as the *A. flavus* AF70 strain. In the *aflU* (*cypA*)–*aflF* (*norB*) region, *A. parasiticus*, which produces G-type AF, retains the complete sequence (Figure 4g, reproduced from Ehrlich et al. [31]). On the other hand, *A. flavus* S-strain and *A. oryzae* have a 1.5 kb deletion in the shared promoter region, and *A. flavus* L-strain has a 0.8 kb deletion in the promoter region and a 32 bp deletion in the *aflF* (*norB*) coding region (Figure 4g). For the coding regions of *aflU* (*cypA*) and *aflF* (*norB*), the annotation of the JCVI-afl1-v3.0 appeared to be inaccurate; therefore, mapping and variant calling were performed against the *A. flavus* NRRL3357 genome assembly JCVI-afl1-v2.0 (accession: GCA_000006275.2) as a reference (Figure 4f). Compared to NRRL3357, JUL1, SEP1, and SEP5 harbored a 610 bp deletion and 32 bp insertion in the *aflF* (*norB*) coding region. However, no variants were found in JUL10 in the entire region. Therefore, based on the *aflU* (*cypA*)–*aflF* (*norB*) region, JUL10 belongs to the L-strain like *A. flavus* NRRL3357, whereas JUL1 belongs to the S-strain.

### 3.5. AF Production Was Not Observed with the Addition of ST in SEP1 and SEP5

In AF BGC, the genes with variants that appeared to have a critical effect on gene function detected only in SEPs were *aflW* (*moxY*), *aflO* (*omtB*), and *aflT*. Both *aflO* (*omtB*) and *aflW* (*moxY*) are involved in the AF biosynthesis pathway prior to ST biosynthesis (Figure 1), and therefore, if *aflP* (*omtA*) and *aflQ* (*ordA*) act normally in SEP1 and SEP5, it is possible that the addition of ST restores AF production in SEP1 and SEP5. ST was added to each strain, and the amount of AF produced in the culture supernatant was quantified (Figure 5a). A significant increase in AF in JUL1 was observed, confirming that the added ST was incorporated into the AF biosynthesis; however, AF production was not observed in SEP1 and SEP5. This indicates that the later steps of the AF biosynthesis, namely *aflP* (*omtA*) and *aflQ* (*ordA*) genes, their transcripts, or their encoding proteins, do not function properly in SEP1 or SEP5. No significant differences were observed in the fungal dry weights of each strain (Figure 5b).

### 3.6. Expression of Several Genes in AF BGC Was Greater in AF-Producing Strains, Especially JUL10

To investigate the reason for the difference in the amount of AF production between NRRL3357, JUL1, JUL10, SEP1, and SEP5, RNA was extracted from each strain after 24, 36, and 48 h of cultivation, and the mRNA levels of the AF BGC genes were quantified by RT-qPCR (Figure 6a). The primers for RT-qPCR were carefully designed to recognize the conserved sequences in the five strains and to check the melting curve of the PCR product to confirm that the target region was amplified in all the strains. At 24 h, around the beginning of AF production, JUL1 and JUL10 showed higher mRNA levels of *aflP* (*omtA*) and *aflC* (*pksA*), whereas no significant differences (>2-fold) were observed for the other genes. At 36 h, when the AF biosynthesis seemed to be most active under our cultivation conditions, the mRNA levels of the enzyme genes for the middle to late steps of the AF biosynthesis, especially *aflD* (*nor-1*), *aflJ* (*estA*), *aflK* (*vbs*), *aflM* (*ver-1*), *aflP* (*omtA*), and *aflQ* (*ordA*), were significantly higher in JUL10 than in the other strains. NRRL3357 showed significantly higher mRNA levels of *aflB* (*fas-1*), *aflC* (*pksA*), and *aflG* (*avnA*) than the non-AF-producing strains. At 48 h, there was no remarkable difference between the AF-producing and AF-non-producing strains for most genes, suggesting the low-level steady expression of cluster genes.

A principal component analysis (PCA) was performed on the RT-qPCR data, and biplots are displayed for the principal components (PC) 1 and PC2 (Figure 6b). The proportions of variance for PC1 and PC2 were 58.4% and 14.9%, respectively, which explained 73.3% of the total variance. In the biplot, the PC1 axis roughly divided the AF-producing and non-producing strains (sample dots) into positive and negative strains; thus, the positive direction of PC1 can be interpreted as AF productivity. In contrast, the PC2 axis appeared to reflect slight differences in expression trends among the genes. Although almost all the biosynthetic genes (vectors) were positively directed along the PC1 axis, only *aflR* was negatively directed. Unexpectedly, *aflR* was expressed at similar levels in all the strains in this analysis, suggesting that, unlike biosynthetic genes, *aflR* expression remained unchanged regardless of AF productivity; hence, aflR was not positively oriented on the PC1 axis.

## 4. Discussion

The JUL1, JUL10, SEP1, and SEP5 strains were initially identified as *A. flavus* based on the colony morphology and the sequence of the ITS region and partial calmodulin gene. However, the genome-wide SNV-based phylogenetic analysis and detailed sequence analysis of AF BGC, especially the *aflR*-*aflS* region, identified SEP1 and SEP5 as *A. oryzae*. Furthermore, based on the reports that the structure of the *aflF* (*norB*)-*aflU* (*cypA*) region is associated with sclerotial morphology in *A. flavus* [31,40], JUL1 was assigned to the *A. flavus* S-type (small sclerotia) strain and JUL10 was assigned to the *A. flavus* L-type (large sclerotia) strain, similar to NRRL3357. However, in the phylogenetic tree based on SNVs, both JUL1 and JUL10 were more closely related to each other and to NRRL3357 than to AF70, the S-strain (Figure 2). This suggests that the difference between S- and L- strains is caused by specific narrow regions including the *aflF* (*norB*)-*aflU* (*cypA*) region, rather than genome-wide differences. Regarding AF production, higher AFB_1_ production in S-strains than in L-strains has been reported [40,41]; however, JUL10 (putative L-strain) produced higher concentrations of AFB_1_ in PDB than JUL1 (putative S-strain), suggesting that AF-producing properties cannot be determined by the *aflF* (*norB*)-*aflU* (*cypA*) region alone. We plan to conduct further experiments which investigate the causal relationship/link/interplay between AF production and sclerotia formation in these strains.

*A. minisclerotigenes* was reported to be closer relatives to *A. oryzae* than *A. flavus* [10], but NRRL3357 was closer to RIB40 in SNV analysis (Figure 2). This difference may be caused by the construction basis of the tree, namely SNVs in the whole genome versus 200 monocore genes (a single homolog gene in each of the targeted species). *A. minisclerotigenes* is considered close to *A. oryzae* RIB40, SEP1, and SEP5 based on monocore gene sequences.

To determine the differences in AF productivity, we focused on the genes with HIGH impact variants and performed GO enrichment analysis for the genes unique to each strain (Figure 3). In JUL1, the sulfur compound metabolic process was enriched, and the *S*-adenosylmethionine synthase gene was found to have a stop-gain variant. Because *S*-adenosylmethionine is used for DMST and ST methylation in AF biosynthesis [42], the expected decrease in *S*-adenosylmethionine may be related to the lower level of AF production in JUL1 than in JUL10. In JUL10, glycerophospholipid catabolic processes and protein phosphorylation are enriched. The enrichment of the glycerophospholipid catabolic process was due to a frameshift variant in the glycerophosphocholine phosphodiesterase gene; however, the relationship between this gene and AF production has not been reported. On protein phosphorylation, a study comparing AF-producing and non-producing *A. parasiticus* showed that the total protein phosphorylation level decreased only during the AF production phase in the AF-producing strain, suggesting that the dephosphorylation of the proteins involved in AF production is required for the onset of AF biosynthesis [43]. Therefore, the expected impairment of protein phosphorylation may result in high AF production in JUL10. The SEP1 and SEP5 strains harbored an intron retention variant in the glycerol-3-phosphate dehydrogenase gene, and the glycerol-3-phosphate metabolic process was enriched. Glycerol-3-phosphate dehydrogenase generates glycerol-3-phosphate from the glycolytic intermediate dihydroxyacetone phosphate. Its gene expression is decreased by dimethylformamide, which inhibits AF production [44]. Further investigation of the relationship between the loss of function of this gene and AF productivity is necessary.

RT-qPCR results confirmed higher expressions of *aflD* (*nor-1*), *aflJ* (*estA*), *aflP* (*omtA*), and *aflQ* (*ordA*) in JUL10 during the AF biosynthesis phase, which may account for the high production of AF in JUL10 (Figure 6). Between JUL1 and NRRL3357 strains, even though AF production was higher in JUL1, the expression of *aflB* (*fas-1*), *aflC* (*pksA*), and *aflG* (*avnA*) was higher in NRRL3357 at 36 h, suggesting that factors outside the AF BGC were related to the differences in AF production. Regarding the gene expression in *A. oryzae* RIB40, many reports have stated little or no expression of *aflR*, *aflJ* (*aflS*), and *aflT*, and no expression of *aflG* (*avnA*), *aflK* (*vbs*), *aflL* (*verB*), and *aflP* (*omtA*) [34,45,46]. On the other hand, Wang et al. reported that mRNA of not only *aflR* but also *aflT*, *aflC* (*pksA*), *aflD* (*nor-1*), *aflA* (*fas-2*), *aflB* (*fas-1*), *aflS* (*aflJ*), *aflH* (*adhA*), *aflJ* (*estA*), *aflE* (*norA*), *aflM* (*ver-1*), *aflP* (*omtA*), and *aflY* (*hypA*) were detected in *A. oryzae* RIB40 by RNA-seq analysis, and stated that the reason for the lack of AF biosynthesis was unclear [47]. Also, in SEP1 and SEP5, most genes in AF BGC, including *aflR*, were expressed, which could not explain the complete absence of AF production. However, the PCA of gene expression data indicated that SEPs were distinct from *A. flavus* NRRL3357 and JULs; that is, the total gene expression levels were lower in SEPs than in AF-producing strains. Because SEPs have stop-gain variants in *aflW* (*moxY*) and intron variants in *aflO* (*omtB*), we hypothesized that ST rescued AF biosynthesis; however, AF production was not detected with ST addition (Figure 5). SEPs also have multiple missense mutations with MODERATE impacts in the genes involved in AF biosynthesis from ST; for example, *aflQ* (*ordA*) has a Pro-29 to Val-29 mutation, any of which could be critical to enzymatic activity [48]. SEPs also contain frameshift variants and deletions in the *aflT* gene, a putative major facilitator superfamily transporter. Although *aflT* does not contribute to AF export, because AF production is not lost in *A. parasiticus aflT* mutants [49], the finding that AflT resides in the aflatoxisome, a specialized trafficking vesicle involved in AF exocytosis, suggests that AflT plays a role in pumping out AF-containing vesicles during the initial stage of AF production [50]. It is also possible that AflT function differs among *A. parasiticus*, *A. flavus*, and *A. oryzae*. Further investigation into whether aflatoxisome-like vesicles are present in SEPs, and into what biosynthetic intermediates are present in SEPs may provide insights into the reasons for AF non-productivity.

In conclusion, the genome analysis in this study could not identify the factors that discriminate AF productivity and non-productivity, but provided clues as to the regulatory factors of AF productivity, including the presence or absence of specific vesicles for AF production and possible metabolomic fluctuation such as glycerol-3-phosphate. Furthermore, our results suggest that *A. flavus* putative S-strain, putative L-strain, and *A. oryzae* can coexist at the same geometric location, and indicate the necessity to proceed with the distribution surveys of *A. flavus* and the genomic analysis of isolates in parallel. Such a comprehensive study would determine the geographic origin of *A. flavus* in Japan. We are currently attempting to isolate AF-producing fungi from soils all over Japan and analyze their genomes, which we plan to report in the future.

## Data Availability

Raw sequencing data are available from the Sequence Read Archive (DRA accession numbers: DRR528608, DRR528609, DRR528610, and DRR528611; https://www.ncbi.nlm.nih.gov/sra; last accessed on 14 June 2024). Other raw data supporting the conclusions of this study will be made available from the authors upon request.

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
