# Peer review of "Comparative Genome Analysis of Japanese Field-Isolated Aspergillus for Aflatoxin Productivity and Non-Productivity"

_jof, 2024, doi:10.3390/jof10070459_

Round 1
Reviewer 1 Report
Aflatoxins are secondary metabolites produced by some Aspergillus fungi and act as potent agricultural toxins causing significant economic losses. Furukawa and co-authors isolate aflatoxin-producing strains from field soils, perform phylogenetic analysis, and determine the production of AF in different Aspergillus fungi. The authors expect that this work will eventually provide information for predicting the potential risk of aflatoxin contamination in agricultural products. However, the content in this work right now is not sufficient to support the conclusion. The manuscript should be reorganized.
Figure 2: The phylogenetic tree should include more Aspergillus species and the output. Besides, the figure legend is not clear.
Figure 4 contains many figures of basic bioinformatics analysis, but Figure 4 doesn't give too much information. Some of the figure should be moved to supplemental data.
GO enrichment analysis in 3.3 indicate several genes outside the aflR-aflS region are necessary to the production of AF.
The author should explain the function of using streigmatocystin to treat Aspergillus fungi.
Author Response
Thank you very much for taking the time to review our manuscript. Please find the detailed responses below and the corresponding revisions (with changes highlighted) in the re-submitted files.
No.1: The biosynthetic origin and mechanism of aflatoxins in the literature should be included in the introduction
Response:
Thank you for your indication. We have added the following sentence in lines 40-42 in the Introduction, “AF is biosynthesized in a multi-enzymatic process consisting of at least 18 steps from acetyl-CoA as a starter unit and Sterigmatocystin (ST) as a penultimate precursor [5] (Scheme 1).”
Supplementary Figure 3, which showed aflatoxin biosynthetic pathway, has been moved to Scheme 1.
No.2: Aflatoxins are secondary metabolites produced by some Aspergillus fungi and act as potent agricultural toxins causing significant economic losses. Furukawa and co-authors isolate aflatoxin-producing strains from field soils, perform phylogenetic analysis, and determine the production of AF in different Aspergillus fungi. The authors expect that this work will eventually provide information for predicting the potential risk of aflatoxin contamination in agricultural products. However, the content in this work right now is not sufficient to support the conclusion. The manuscript should be reorganized.
Response:
Thank you very much for accurately summarizing our manuscript.
The purpose of this study is to identify the phylogenetic position of A. flavus isolates from soil of Ibaraki prefecture , where aflatoxin-producing fungi have been thought to be absent, and to investigate the factors discriminating AF production and nonproduction from genomic analysis. Predicting the potential risk of aflatoxin contamination in agricultural products is the important goal of research on AF, but it is beyond the scope of this study.
As you pointed out, we have not been able to confirmatively identify the factors responsible for AF production and non-production. However, we think this study is worth reporting since it provided meaningful data that fungal strains with diversity in terms of phylogeny and AF-productivity can be isolated from geographically identical locations, as well as data on genomic differences related to AF-productivity.
We are currently isolating AF-producing fungi from soil samples from all over Japan and attempting to analyze their genome, which we plan to report in the future.
No. 2: Figure 2: The phylogenetic tree should include more Aspergillus species and the output. Besides, the figure legend is not clear.
Response:
Thank you for your valuable suggestions. In addition to six strains (the four isolates, A. oryzae RIB40, and A. flavus NRRL3357), we downloaded sequencing data of A. flavus AF70 strain, A. parasiticus CBS 117618 strain, and A. minisclerotigenes CBS 117635 strain, which was recently reported to be close relatives to A. oryzae, and performed phylogenetic analysis. Since this study aims to determine the phylogenetic position of the four isolates rather than explore the phylogenetic relationships among variety of Aspergillus species, we think that nine strains are sufficient. Fig. 2 has been replaced with the new data and the relevant sentences in Materials and Methods, Results, and Discussion has been revised. We have also rewritten the legend of Fig. 2 to be more detailed.
No. 3: Figure 4 contains many figures of basic bioinformatics analysis, but Figure 4 doesn't give too much information. Some of the figure should be moved to supplemental data
Response:
Thank you for your indication. We moved aflY (nadA), which is not related to AFB1 biosynthesis, aflLa (hypB), whose function is not well defined, and aflL (verB), which has been predicted to have frameshift mutation in all four strains, to Fig, S3. Accodingly, Fig. 2 has 2 pages now.
No. 4: GO enrichment analysis in 3.3 indicate several genes outside the aflR-aflS region are necessary to the production of AF.
Response:
Thank you for your comment. As you pointed out, genes located outside the AF biosynthesis gene cluster are required for AF production. Although almost all of biosynthetic enzymes are clarified and located in gene cluster, there are many unclarified points in the regulatory mechanism of the gene expression of the transcription factors aflR and aflS themselves. Factors such as carbon source, pH, and nitrogen source are known to be involved in the regulation of aflR expression.
Our genomic and gene expression analyses indicate that two strains (SEP1 and SEP5) do not produce AF even when aflR and other AF biosynthetic genes are expressed. We think that some of the genes found in the current GO analysis may be responsible for the non-production of AF, and we plan to analyze them in more detail in the future.
No. 5: The author should explain the function of using sterigmatocystin to treat Aspergillus fungi.
Response:
Thank you for your remarks. We are sorry that the meaning of treatment of sterigmatocystin was not sufficiently explained.
Sterigmatocystin (ST) is the penultimate precursor of AF biosynthesis, and AFB1 is biosynthesized from ST by aflP (omtA) and aflQ (ordA). Because there are no impact HIGH variants in the aflP (omtA) and aflQ (ordA) gene in SEP1 and SEP5, we hypothesized that the addition of ST would rescue AF production in SEP1 and SEP5.
Since AF production was increased in JUL1 with the addition of ST, ST was supposed to be incorporated into fungal cells. However, in SEP1 and SEP5, AF production was not restored, suggesting that either ST was not incorporated or that aflP (omtA) and aflQ (ordA) were not functioning properly.
For clarity, Scheme 1 was added.
We changed the sentence in lines 421-424 in the section 3.5., to “Both aflO (omtB) and aflW (moxY) are involved in the AF biosynthesis pathway prior to ST biosynthesis (Scheme 1), and therefore if aflP (omtA) and aflQ (ordA) act normally in SEP1 and SEP5, it is possible that the addition of ST restores AF production in SEP1 and SEP5.”
Reviewer 2 Report
The paper aims to identify differences between aflatoxin-producing and non-producing strains. For this purpose the authors isolated four Aspergillus strains from agricultural fields and characterized and sequenced them. Two strains turned out to produce aflatoxin whereas two did not synthesize the toxin.
Since aflatoxin contaminates various food products and is a potent carcinogen it is of high relevance how the taxonomically closely related strains can be easily distinguished. Also their distribution in the environment and under different growth conditions is of great interest.
The paper reports a huge amount of sequence data and sequence comparisons as well as expression studies. But unfortunately, at the end, there is no real clue what makes a strain a producer or a non-producer. Also no clear correlation between other criteria (such as origin of strain, growth behaviour, etc.) and toxin production were detected.
It is a little bit frustrating (for the reader, and probably for the authors too) that there is a great number of primary observations, but no conclusive explanation for the differences in the ability to synthesize aflatoxin. But this study may be a good starting point for further analyses.
There are a few general points that should be noted.
- It is difficult to follow the rationale of some studies (e.g. feeding experiments) if one is not familiar with the specific aflatoxin biosynthetic pathway. A scheme would be very helpful. The link to the Supplementary Material did not work. Anyhow, I would suggest to include the scheme in the Introduction.
- The isolation of the strains is reported in the last paragraph of the Introduction and in Materials & Methods. I recommend to describe this in a first chapter of the Results (with experimental details in M&M).
- Some more information on the criteria how the four different strains were selected would be helpful.
-
|
Line |
Special Comment |
|
2-3 |
Production versus Productivity? è Alternative: Comparative genome analysis of Japanese field isolated Aspergillus for aflatoxin production and non-production |
|
56-57 |
This sentence might be misleading. It sounds like it is the aim to find AF-producing strains. è Alternative: To determine the potential risk of AF contamination in local agricultural products, it is important to investigate the geological distribution AF-producing strains in field soils. |
|
69 |
Replacing “putative A. flavus strains” by “Aspergillus strains” |
|
70 |
Is there a particular reason for isolating Aspergillus strains in July and September? If so, please provide a justification. |
|
72-73 |
Grammar/language: “The ITS region sequences were analyzed to identify A. flavus in these strains.” è Alternative: Sequencing of the ITS regions allowed the identification of these strains as A. flavus. |
|
81 |
Language: „oberbation“ è observation |
|
85 |
Language: “litter” è liter or litre |
|
86 |
Language: “litter” è liter or litre |
|
87 |
Some extra information for cultivation on plate might be useful. è e. g. Temperature, light/dark |
|
87 |
Technical language: “… conidia were inoculated into potato dextrose broth … “ è For AFB1 analysis, potato dextrose broth was inoculated with conidia … |
|
92 |
What does AFB1 staining mean? è According to the manuscript they were analyzed by LC-MS, not by staining. |
|
104 - 107 |
LC-MS is not an optimal method for quantification without labeled aflatoxin standards. Better way to do it: 1. Checking for production by LC-MS (EIC: m/z = 313.0691 – 313.0723) 2. If production can be observed: Quantification of concentrated extracts using the absorption of 365 nm and/or 455 nm for a specific aflatoxin derivative (which can be compared to a known internal standard using HPLC or UHPLC). However, a calibration curve might be sufficient for this kind of measurements. |
|
109 |
PDB = Potato dextrose broth à „PDB liquid medium“ is somehow redundant |
|
193 |
Incubation for 7 days at which temperature? |
|
197-198 |
Alternative: In NRRL3357, JUL1 and JUL10 AFB1 was detected after merely 24 hours. Within the following 12 hours, a relatively strong increase in AFB1 production was observed for the two isolates JUL1 and JUL10 in contrast to NRRL3357. After 48 hours, AFB1 accumulation was 12 ppb for NRRL3357, 206 ppb for JUL1 and 792 ppb for JUL10. |
|
228 |
This sentence might be hard to understand because of its structure. JUL1 and JUL10 similar to NRRL3357 à AF producer SEP1 and SEP5 similar to RIB40 à AF non-producer è The tree show that while JUL1 and JUL10 were close to AF-producing strain A. flavus NRRL3357, SEP1 and SEP5 were close to the AF-non-producing strain A. oryzae RIB40. |
|
240 |
Examples for LOW and MODIFIER? |
|
242 |
Stop-gain = non-sense mutation? |
|
243 |
“… amino acid mutations, such as missense mutations, are classified as MODERATE.” è This view might be too oversimplified. Missense mutations in the active center of an enzyme should be considered as HIGH whereas mutation in enzymes concerning for example the C-terminal end might be only MODERATE. |
|
282 |
Genetic polymorphisms = Genetic variants? SNP is not equal to SNV (according to literature) |
|
321 |
AflR and AflS (AflJ) = proteins à Capital letters |
|
336 - 341 |
Do you mean genes or proteins? (see: 321) |
|
359 |
Frameshift variant at proline 19? Frameshift à DNA not protein level? |
|
360 |
AflL = protein à Capital letters |
|
364 |
aflU (cypA) function or AflU (CypA) function à Capital letters à Is “functionality” connected to proteins or to genes? |
|
365ff |
S-type (small sclerotia) versus L-type (large sclerotia) à Connection to AF production is unclear because this concept is neither mentioned before nor introduced at any previous sentences. More background is needed at this point. è Information from line 372 should be mentioned here: NRRL3357 = L-type strain = AF producer è Is sclerotia formation (= morphology) linked to AF production? No! JUL1 (S-type) and JUL10 (L-type) are both AF producer.
Why this additional discrimination of S-type (small sclerotia) versus L-type (large sclerotia) is introduced? The significance of this distinction is incomprehensible (for me). |
|
381 |
… and the addition of ST rescues AF production in SEP1 and SEP5 è This partial sentence does not fit at all (to the figure): è Have JUL and SEP strains been mixed up? The figure 5) shows the opposite. |
|
385-386 |
This sentence suggests that SEP1 and SEP5 are enzymes within a biosynthetic pathway. è This sentence does not make sense at all (for me) |
|
386 |
… no significant differences … (sounds a little bit better in my opinion) |
|
392 |
Same headline as in line 377 |
|
420, Fig 6a |
Explain the meaning of “a”, “b” etc. in the legend |
|
433 |
Why calmodulin gene? Are ITS regions not enough? If so, why? |
|
436 |
This information should be mentioned in results part, see: 365ff |
|
437 |
Does this mean that AF production is correlated with morphology? If so, one should mention this possible correlation before. |
|
446 |
Better: “We plan to conduct further experiments which investigate the causal relationship/link/interplay between AF production and sclerotia formation in these strains. |
|
498 |
aflT à AflT (= protein) à Capital letters |

Author Response
Thank you very much for taking the time to review our manuscript. Please find the detailed responses below and the corresponding revisions (with changes highlighted) in the re-submitted files.
No.1: The paper aims to identify differences between aflatoxin-producing and non-producing strains. For this purpose the authors isolated four Aspergillus strains from agricultural fields and characterized and sequenced them. Two strains turned out to produce aflatoxin whereas two did not synthesize the toxin.
Since aflatoxin contaminates various food products and is a potent carcinogen it is of high relevance how the taxonomically closely related strains can be easily distinguished. Also their distribution in the environment and under different growth conditions is of great interest.
The paper reports a huge amount of sequence data and sequence comparisons as well as expression studies. But unfortunately, at the end, there is no real clue what makes a strain a producer or a non-producer. Also no clear correlation between other criteria (such as origin of strain, growth behaviour, etc.) and toxin production were detected.
It is a little bit frustrating (for the reader, and probably for the authors too) that there is a great number of primary observations, but no conclusive explanation for the differences in the ability to synthesize aflatoxin. But this study may be a good starting point for further analyses.
Response:
Thank you very much for accurately summarizing our manuscript and for recognizing the value of our study. Your point is very perceptive. Following this study, we are planning a phylogenetic analysis of A. flavus isolates from fields all over Japan and will report it in the future and try to find the difference between AF producer and non-producer.
No.2: - It is difficult to follow the rationale of some studies (e.g. feeding experiments) if one is not familiar with the specific aflatoxin biosynthetic pathway. A scheme would be very helpful. The link to the Supplementary Material did not work. Anyhow, I would suggest to include the scheme in the Introduction.
Response:
Thank you for your helpful remarks. We have not taken into consideration those who are not familiar with aflatoxin biosynthesis.
We changed the aflatoxin biosynthesis pathway in Supplementary figures to Scheme 1 in the Introduction. We added the following sentence to lines 40-42 in the Introduction.
“AF is biosynthesized in a multi-enzymatic process consisting of at least 18 steps from acetyl-CoA as a starter unit and Sterigmatocystin (ST) as a penultimate precursor [5] (Scheme 1).”
No.3: - The isolation of the strains is reported in the last paragraph of the Introduction and in Materials & Methods. I recommend to describe this in a first chapter of the Results (with experimental details in M&M).
Response:
Thank you for your good suggestion. We did not write about the isolation of the four strains because it has already been reported in Reference 14. However, since Reference 14 is written in Japanese, it would be more helpful to write more details in the Materials and Methods section, as you mentioned.
We added the following sentences in lines 87-92, in the section 2.1.
“In detail, 0.15 g of soil 5 cm below the surface was collected and diluted in 500 μL of 0.05% tween-20, then spread on YES agar plates (20 g yeast extract (Difco, Sparks, MD, USA), 20 g agar, 100 g sucrose, 1 g sodium deoxycholate, 0.1 g chloramphenicol per liter) and incubated at 25 °C in the dark for 7 days. Colonies that were visually similar to A. flavus were isolated. From these isolates, sequencing of the ITS regions and partial calmodulin gene allowed the identification of four strains as A. flavus.”
At the beginning of the results in lines 218-220, we added “In 2020, four putative Aspergillus strains were isolated from a field soil in Ibaraki, Japan, and named JUL1, JUL10, SEP1, and SEP5. These four strains were judged to be A. flavus based on colony morphology and DNA barcode region.”
No.4: - Some more information on the criteria how the four different strains were selected would be helpful.
Response:
Thank you for your indication. As noted in the answer to No. 3, we added the following sentences in lines 90-92 in the section 2.1., “Colonies that were visually similar to A. flavus were isolated. From these isolates, sequencing of the ITS regions and partial calmodulin gene allowed the identification of four strains as A. flavus.”
No. 5: Production versus Productivity? âž” Alternative: Comparative genome analysis of Japanese field isolated Aspergillus for aflatoxin production and non-production
Response:
Thank you for your suggestion. The title of Reference 32 is "Aflatoxin non-productivity of Aspergillus oryzae caused by loss of function in the aflJ gene product”. We will follow this and leave productivity and non-productivity as they are.
No.6: This sentence might be misleading. It sounds like it is the aim to find AF-producing strains. âž” Alternative: To determine the potential risk of AF contamination in local agricultural products, it is important to investigate the geological distribution AF-producing strains in field soils.
Response:
Thank you your to the point text revision. We have corrected (Line 64-65), but we changed “determine” to “estimate”.
No.7: Replacing “putative A. flavus strains” by “Aspergillus strains”
Response:
We have corrected. (Line 85)
No.8: Is there a particular reason for isolating Aspergillus strains in July and September? If so, please provide a justification.
Response:
Thank you for your question. The soil in the field was sampled in July, August, September, and October for no particular reason. There are no A. flavus-like colonies from the soil samples of August and October. We did not pursue the reason.
Therefore, we just randomly selected two colonies from each of July and September for analysis.
No.9: Grammar/language: “The ITS region sequences were analyzed to identify A. flavus in these strains.” âž” Alternative: Sequencing of the ITS regions allowed the identification of these strains as A. flavus.
Response:
We have corrected. (Line 92-93)
No. 10: Language: „oberbation“ âž” observation
Response:
We have corrected. (Line 102)
No. 11: Language: “litter” âž” liter or litre
Response:
We have corrected. (Line 106)
No. 12: Language: “litter” âž” liter or litre
Response:
We have corrected. (Line 107)
No. 13: Some extra information for cultivation on plate might be useful. âž” e. g. Temperature, light/dark
Response:
Thank you very much. We have added “at 28°C in the dark”. (Line 108)
No. 14: Technical language: “… conidia were inoculated into potato dextrose broth … “ âž” For AFB1 analysis, potato dextrose broth was inoculated with conidia …
Response:
We have corrected. (Lines 108-109)
No. 15: What does AFB1 staining mean? âž” According to the manuscript they were analyzed by LC-MS, not by staining.
Response:
Sorry for our mistake. We have corrected “AFB1 staining” to “AFB1 analysis”. (Line 113)
No. 16: LC-MS is not an optimal method for quantification without labeled aflatoxin standards. Better way to do it: 1. Checking for production by LC-MS (EIC: m/z = 313.0691 – 313.0723) 2. If production can be observed: Quantification of concentrated extracts using the absorption of 365 nm and/or 455 nm for a specific aflatoxin derivative (which can be compared to a known internal standard using HPLC or UHPLC). However, a calibration curve might be sufficient for this kind of measurements.
Response:
Thank you for your instruction regarding the optimal aflatoxin analysis method. As you indicated, the method of detecting and quantifying derivatized aflatoxin by fluorescence is accurate. However, we think that LC/MS analysis using an aflatoxin standard dilution series for making a calibration curve is sufficient for this study.
No. 17: PDB = Potato dextrose broth → „PDB liquid medium“ is somehow redundant
Response:
Thank you very much. We have changed “PDB liquid medium” to “PDB” (Line 134, and other places)
No. 18: Incubation for 7 days at which temperature?
Response:
Thank you very much. We have added “at 28°C”. (Line 221)
No. 19: Alternative: In NRRL3357, JUL1 and JUL10 AFB1 was detected after merely 24 hours. Within the following 12 hours, a relatively strong increase in AFB1 production was observed for the two isolates JUL1 and JUL10 in contrast to NRRL3357. After 48 hours, AFB1 accumulation was 12 ppb for NRRL3357, 206 ppb for JUL1 and 792 ppb for JUL10.
Response:
Thanks for to the point text revision. We have revised it as you suggested. (Lines 224-228)
No. 20: This sentence might be hard to understand because of its structure. JUL1 and JUL10 similar to NRRL3357 → AF producer SEP1 and SEP5 similar to RIB40 → AF non-producer âž” The tree show that while JUL1 and JUL10 were close to AF-producing strain A. flavus NRRL3357, SEP1 and SEP5 were close to the AF-non-producing strain A. oryzae RIB40.
Response:
Thanks for to the point text revision. We have revised it as you suggested. (Lines 256-258)
No. 21: Examples for LOW and MODIFIER?
Response:
Thank you for your remarks. We have added the following sentences to section 3.3 (Lines 278-281). “Variants unlikely to change protein function, such as synonymous variant (codon change that produces the same amino acid) are classified as LOW. Variants in non-coding region where there is no evidence of impact, such as variants in downstream of a gene are classified as MODIFIER.”
No. 22: Stop-gain = non-sense mutation?
Response:
As you mentioned, stop-gain variant is identical to nonsense variant. Stop-gain variant is a variant that changes at least one base of a codon, leading to a premature stop codon.
No. 23: ” … amino mutations, such as missense mutations, are classified as MODERATE.” âž” This view might be too oversimplified. Missense mutations in the active center of an enzyme should be considered as HIGH whereas mutation in enzymes concerning for example the C-terminal end might be only MODERATE.
Response:
Your remark is to the point. Variants that result in amino acid changes close to the active site may be fatal to protein function. However, in the categorization by snpEff, location of the variants in the protein structure is not considered.
I have added the following text to lines 277-278 of the Results.
“Depending on location, MODERATE variants can be fatal to protein function, but are not judged by SnpEff.”
No. 24: Genetic polymorphisms = Genetic variants? SNP is not equal to SNV (according to literature)
Response:
Sorry for the improper wording. As you said, polymorphism differs from variants, in that we need to investigate the frequency in the population. We corrected it to “Genetic variants”. (Line 321)
No. 25: AflR and AflS (AflJ) = proteins → Capital letters
Response:
Thank you for pointing this out. We have corrected it to lowercase italics. (Line 352)
No. 26: Do you mean genes or proteins? (see: 321)
Response:
Sorry for the inappropriate sentence. We have rewritten it as follows.
“aflYa (nadA) is presumed to encode the enzyme involved in the final step of AFG1 and AFG2 biosynthesis together with the enzymes encoded by aflF (norB) and aflU (cypA) [35]” (Line 374-375)
No. 27: Frameshift variant at proline 19? Frameshift → DNA not protein level?
Response:
Sorry for the inappropriate wording. We have changed to “frameshift variant at codon 19 (Proline 19 frameshift)” (Line 398)
Similarly, we have changed “at residue 99” to “at codon 99” in line 376.
No. 28: AflL = protein → Capital letters
Response:
Thank you for your indication. We have changed to “AflL (VerB)” (Line 399)
No. 29: aflU (cypA) function or AflU (CypA) function → Capital letters → Is “functionality” connected to proteins or to genes?
Response:
Thank you for your indication. We have changed to “AflU (CypA)” (Line 0403)
Here we mentioned protein function of AflU (CypA).
No. 30: S-type (small sclerotia) versus L-type (large sclerotia) → Connection to AF production is unclear because this concept is neither mentioned before nor introduced at any previous sentences. More background is needed at this point. âž” Information from line 372 should be mentioned here: NRRL3357 = L-type strain = AF producer âž” Is sclerotia formation (= morphology) linked to AF production? No! JUL1 (S-type) and JUL10 (L-type) are both AF producer. Why this additional discrimination of S-type (small sclerotia) versus L-type (large sclerotia) is introduced? The significance of this distinction is incomprehensible (for me).
Response:
We are sorry for the confusion because of poor sequence of sentences and lack of explanation.
We added following sentences in lines 403-407,
“The structure of the aflU (cypA)–aflF (norB) region has been reported to correlate with AF production levels and sclerotial morphology in some strains of A. flavus [31, 40]. A. flavus NRRL3357 is a large sclerotial type (L-strain) and is thought to produce less AF than small sclerotial type (S-strain) such as A. flavus AF70 strain.”
Reference 40 contains the following statement:
“It has been reported that the inability of A. flavus to produce G-aflatoxins is based on partial
or complete deletion of the aflU gene, and that the amount of deletion observed in the aflF/ aflU region will correlate with sclerotial morphology [59]. For example, the L-strain genotype for the aflF/aflU region results in an amplicon that is approximately 1 kb in size, while the S-strain genotype results in amplicon size of approximately 300 bp. Therefore, based on the report of Ehrlich and co-workers [59], NRRL 3357 is an L-morphotype strain and AF70 is an S-morphotype strain,”
and
” For example, the S-morphotypes produces sclerotia that are smaller (< 400 μm), greater in quantity, and contain higher concentrations of aflatoxin than the L-morphotypes (>400 μm)”
Reference 40: Gilbert MK, Mack BM, Moore GG, et al. Whole genome comparison of Aspergillus flavus L-morphotype strain NRRL 3357 (type) and S-morphotype strain AF70. PLoS One. 2018;13(7):1-16. doi:10.1371/journal.pone.0199169
No. 31: … and the addition of ST rescues AF production in SEP1 and SEP5 âž” This partial sentence does not fit at all (to the figure): âž” Have JUL and SEP strains been mixed up? The figure 5) shows the opposite.
Response:
Sorry for the confusion in the sentence. It should have been a hypothetical statement.
JUL1, JUL10, SEP1, SEP5, and NRRL3357 are cultured separately.
I have rewritten the sentence as follows.
“Both aflO (omtB) and aflW (moxY) are involved in the AF biosynthesis pathway prior to ST biosynthesis (Scheme 1), and therefore if aflP (omtA) and aflQ (ordA) act normally in SEP1 and SEP5, it is possible that the addition of ST restores AF production in SEP1 and SEP5. ” (Line 422-425)
No. 32: This sentence suggests that SEP1 and SEP5 are enzymes within a biosynthetic pathway. âž” This sentence does not make sense at all (for me)
Response:
We are sorry for the sentence that doesn't make sense. We modified the sentence to “This indicates that the later steps of AF biosynthesis, namely aflP (omtA) and aflQ (ordA) genes, their transcripts, or their encoding proteins, do not function properly in SEP1 or SEP5.” (Line429-431)
No. 33: 386 … no significant differences … (sounds a little bit better in my opinion)
Response:
Thank you for your suggestion. We added “significant” as you mentioned. (Line 431)
No. 34: 392 Same headline as in line 377
Response:
Sorry for our mistake. We changed the subheading to “3.6 Expression of several genes in AF BGC was greater in AF-producing strains, especially JUL10.”
No. 35: 420, Fig 6a Explain the meaning of “a”, “b” etc. in the legend
Response:
The Tukey’s test was used to detect statistical differences, and the groups with significant differences are indicated by different letters.
We have changed the legend to “Statistical differences (p < 0.05) among strains were determined by Tukey’s multiple comparisons test (There are Significant differences between bars of different letters).”
No. 36: 433 Why calmodulin gene? Are ITS regions not enough? If so, why?
Response:
The partial calmodulin gene is used to classify fungi of the genus Aspergillus. While analysis of the ITS region alone only leads to Aspergillus spp., including A. fumigatus, analysis of the calmodulin sequence can narrow it down to A. flavus or A. oryzae.
Ssince the identical sequences are registered for A. flavus and A. oryzae on calmodulin gene, partial calmodulin gene could not discriminate between them.
I added the description about the analysis of calmodulin gene in Materials and Methods (line 93).
Reference:
Hong S-B, Go S-J, Shin H-D, Frisvad JC, Samson RA. Polyphasic taxonomy of Aspergillus fumigatus and related species. Mycologia. 2005;97(6):1316-1329.
No. 37: 436 This information should be mentioned in results part, see: 365ff
Response:
Thank you for your indication. See the response to No. 30.
No. 38: 437 Does this mean that AF production is correlated with morphology? If so, one should mention this possible correlation before.
Response:
Thank you for your indication. See the response to No. 30.
No. 39: 446 Better: “We plan to conduct further experiments which investigate the causal relationship/link/interplay between AF production and sclerotia formation in these strains.
Response:
Thank you for to the point text revision. We have revised it as you suggested. (lines 493-494)
No. 40: 498 aflT → AflT (= protein) → Capital letters
Response:
Thank you for your indication. We changed “aflT” to “AflT” in lines 547 and 549 in revised manuscript.
Reviewer 3 Report
The authors describe of the genomic sequences and gene expressions of the aflatoxin gene cluster of four strains of A. flavus isolated in Japan this decade. The manuscript is well-written and contributes to the literature of known variations in the gene cluster.
I do not have any suggestions for improving the manuscript.
Author Response
Thank you very much for taking the time to review our manuscript. Thank you very much for accurately summarizing our manuscript and for recognizing the value of our study.
Reviewer 4 Report
The present work is a contribution to the study of the diversity of Aspergillus flavus in a particular area of Japan (Tsukuba, Ibaraki), as well as the ability of these strains to produce the mycotoxin aflatoxin B1. The work has been well developed and the authors have studied in great detail the 4 strains under study, using advanced methodology that is well suited for this type of study. However, it would have been interesting to study other strains from other areas of Japan to investigate the distribution throughout the country. As no climate data are analysed, it is not clear how this study contributes to the expected increase of toxigenic aspergillus and aflatoxins due to climate change.
Introduction
Lines 30-31: A mention of Aspergillus parasiticus as another aflatoxin-producing fungus would be adequate
There is a lack of a formal description of what the objectives of this research are.
Material and methods
Line 92: what do you mean by AFB1 staining?
Line 198: Do not use the term ppb, use ng/mL or µg/L instead
Line 377 and 392: The name of subheading 3.5. AF production was not observed with the addition of ST in SEP1 and SEP5 is repeated
Author Response
Thank you very much for taking the time to review our manuscript. Please find the detailed responses below and the corresponding revisions (with changes highlighted) in the re-submitted files.
No. 1: The present work is a contribution to the study of the diversity of Aspergillus flavus in a particular area of Japan (Tsukuba, Ibaraki), as well as the ability of these strains to produce the mycotoxin aflatoxin B1. The work has been well developed and the authors have studied in great detail the 4 strains under study, using advanced methodology that is well suited for this type of study.
Response:
Thank you very much for accurately summarizing and recognizing the value of our study.
No. 2: However, it would have been interesting to study other strains from other areas of Japan to investigate the distribution throughout the country.
Response:
Thank you for your meaningful remarks. In fact, we are in the process of isolating AF-producing fungi from field soils throughout Japan, and are planning to conduct comparative genomic analysis of the isolates. We will report the results in the future. In this sense, this study is incomplete, but we think that this study is worth reporting since it provided meaningful data that fungal strains with diversity in terms of phylogeny and AF-productivity can be isolated from geographically identical locations, as well as data on genomic differences related to AF-productivity.
No. 3: As no climate data are analysed, it is not clear how this study contributes to the expected increase of toxigenic aspergillus and aflatoxins due to climate change.
Response:
Thank you for your helpful comments. We think that studies connecting climate change to the spread of AF-producing fungi and AF contamination are beyond the scope of this paper because they require elaborately programmed simulations.
However, it is desirable to mention climate change in the area. So, we have added the following sentence to Lines 70-73, “In fact, according to the Japan Meteorological Agency, the long-term trend of annual average temperatures in Japan and Ibaraki have been increasing at a rate of 1.35 °C and 2.3 °C per 100 years, respectively; in 2023, the annual average temperature reached 16.1 °C in Tsukuba, Ibaraki.”
No. 4: Lines 30-31: A mention of Aspergillus parasiticus as another aflatoxin-producing fungus would be adequate
Response:
Thank you for your indication. In lines 32-34, we added the following sentence “AF-producing fungi include Aspergillus flavus, A. parasiticus, and the less common A. nomius, A. pseudotamarii, A. bombycis, and A. parvisclerotigenus. ”
No. 5: There is a lack of a formal description of what the objectives of this research are.
Response:
Thank you for your indication. In lines 76-79 in the Introduction, we added the following sentences to indicate the objective of this study, “To date, no study has analyzed the genomes of A. flavus isolated from Japanese soils. Genomic analysis of field isolates is expected to reveal phylogenic characteristic of strains in the area and provide clues to novel factors that discriminate between AF-producing and non-producing strains.”
No. 6: Line 92: what do you mean by AFB1 staining?
Response:
Sorry for our mistake. We have corrected “AFB1 staining” to “AFB1 analysis”.
No. 7: Line 198: Do not use the term ppb, use ng/mL or µg/L instead
Response:
Thank you for your indication. We changed ppb to ng/mL in all places.
No. 8: Line 377 and 392: The name of subheading 3.5. AF production was not observed with the addition of ST in SEP1 and SEP5 is repeated
Response:
Sorry for our mistake. We changed the latter subheading to “3.6 Expression of several genes in AF BGC was greater in AF-producing strains, especially JUL10.”
Round 2
Reviewer 1 Report
The revised manuscript is OK.
The revised manuscript is OK.